# Serum Metabolomics Profiling Reveals Metabolic Alterations Prior to a Diagnosis with Non-Small Cell Lung Cancer among Chinese Community Residents: A Prospective Nested Case-Control Study

**DOI:** 10.3390/metabo12100906

**Published:** 2022-09-27

**Authors:** Yu Xiang, Qi Zhao, Yilin Wu, Xing Liu, Junjie Zhu, Yuting Yu, Xuyan Su, Kelin Xu, Yonggen Jiang, Genming Zhao

**Affiliations:** 1Key Laboratory of Public Health Safety of Ministry of Education, Department of Epidemiology, School of Public Health, Fudan University, Shanghai 200032, China; 2Songjiang District Center for Disease Control and Prevention, Shanghai 201600, China; 3Department of Epidemiology and Health Statistics, School of Public Health, Dali University, Dali 671000, China; 4Department of Biostatistics, School of Public Health, Key Laboratory of Public Health Safety of Ministry of Education, Fudan University, Shanghai 200032, China

**Keywords:** non-small cell lung cancer (NSCLC), metabolomics, cancer cells metabolism, nested case-control study

## Abstract

The present high mortality of lung cancer in China stems mainly from the lack of feasible, non-invasive and early disease detection biomarkers. Serum metabolomics profiling to reveal metabolic alterations could expedite the disease detection process and suggest those patients who are harboring disease. Using a nested case-control design, we applied ultra-high-performance liquid chromatography/mass spectrometry (LC-MS)-based serum metabolomics to reveal the metabolomic alterations and to indicate the presence of non-small cell lung cancer (NSCLC) using serum samples collected prior to disease diagnoses. The studied serum samples were collected from 41 patients before a NSCLC diagnosis (within 3.0 y) and 38 matched the cancer-free controls from the prospective Shanghai Suburban Adult Cohort. The NSCLC patients markedly presented cellular metabolism alterations in serum samples collected prior to their disease diagnoses compared with the cancer-free controls. In total, we identified 18 significantly expressed metabolites whose relative abundance showed either an upward or a downward trend, with most of them being lipid and lipid-like molecules, organic acids, and nitrogen compounds. Choline metabolism in cancer, sphingolipid, and glycerophospholipid metabolism emerged as the significant metabolic disturbance of NSCLC. The metabolites involved in these biological processes may be the distinctive features associated with NSCLC prior to a diagnosis.

## 1. Introduction

Lung cancer is currently the leading cause of death in China [1]. According to the International Agency for Research on Cancer, in 2020, the estimated number of incidents of lung cancer was over three million in China, accounting for nearly 19.4% of deaths in both sexes and all ages [2]. Moreover, the survival rate of lung cancer remains dismal. Most early-stage lung cancers are asymptomatic, contributing to a delayed diagnosis and resulting in an overall 5 y survival rate of 19% over all stages; however, the detection of lung cancer at the early stage could increase the 5 y survival rates, rising to 57% [3]. This significant difference indicates the importance of detecting lung cancer at an early stage for improving the overall survival rate. With the available treatments having increased in recent years, the outcomes with an early detection would also be improved.

At present, low-dose spiral computerized tomography (LDCT) is considered an effective imaging tool for detecting early-stage lung lesions and for screening high-risk populations of lung cancer [4]. However, the screening recommendations have been subjectively assessed based on smoking behavior, resulting in a potential reluctance on the part of patients. Alongside concerns about the economic burden to the nation, the potential radiation hazards and psychological burdens to patients limit the feasibility of widespread LDCT screening [5]. In addition, the false-positive rate in a low-dose group, for example, was estimated to be 96.4%, with most positive results having been withdrawn after further imaging examinations [6]. These reasonable arguments support the urgent need for a less invasive screening test, either without or with less harmful side effects, to provide an alert for the suspicious signs of early neoplasm. Such tests could help medical practitioners triage patients with malignancy, support an advanced treatment efficiency, and improve treatment outcomes to lower lung cancer-associated mortality.

Molecular biology, genomics, proteomics, and metabolomics are interrelated in the probing and identifying of biological processes. Genomics measures genetic mutations and predicts the possibility of disease development at the individual level; however, proteomics and metabolomics could identify ongoing biological activities, including alterations due to the presence of disease [7]. In cancer, metabolomics captures complex physiological and pathological characteristics, and detects oncological developments via measurable metabolic profiles from the metabolic pathways through global metabolite variations.

Although metabolomic studies of cancer are carried out frequently, most of these studies are conducted with case-control study designs, and the results might be prone to reverse causation [8,9,10,11]. Currently, the serum metabolomic profiling revealing metabolic alterations prior to a diagnosis with lung cancer in the Chinese population is scarce and limited. The Shanghai Suburban Adult Cohort and Biobank (SSACB) study is a population-based prospective cohort that included 36,404 participants from the Songjiang district, in Shanghai, China, who were predominantly community residents aged 20–74 years old. Here, we report the metabolic alterations established through blood serum samples collected before the diagnosis of non-small cell lung cancer (NSCLC) and compare the patients’ results with the serum samples obtained from healthy, cancer-free control subjects in the Shanghai Suburban Adult Cohort. Our study provides an opportunity to prospectively investigate the role of metabolites and their alteration in serum samples collected before the diagnosis of NSCLC to explore its associated biological processes to prevent disease precisely. To our knowledge, our study was the first prospective study to investigate the pre-diagnostic serum metabolic biomarkers and pathway alterations with the liquid chromatography/mass spectrometry (LC-MS) technique among the community residents in Shanghai. Our results demonstrate the potential of LC-MS-based serum metabolites in conjunction with baseline characteristics to indicate the presence of NSCLC, and its potential to be used as an effective tool for screening high-risk patients of NSCLC.

## 2. Materials and Methods

### 2.1. Ethics

This study was performed in conformity with the Declaration of Helsinki and was approved by the Institutional Review Board of Fudan University, School of Public Health, China (Authorization number: IRB#2016-04-0586). Informed consent was obtained from all participants.

### 2.2. Design and Subjects

From April 2016 to September 2017, the SSACB study recruited 36,404 participants aged 20–74 years old, primarily community residents living in the Songjiang District, in Shanghai, China. Blood samples of the participants were drawn during the baseline investigation. Detailed descriptions of our participants’ recruitment criteria and phases are illustrated in the cohort profile [12]. Sociodemographic information, health status (self-evaluation), lifestyle, and the history of chronic diseases were collected using a structured questionnaire at the baseline investigation for all the participants included in the SSCAB study. Fasting blood serum samples were collected from the participants enrolled in the baseline investigation. Afterward, the serum samples were transported to the biobank on ice at approximately 0 °C to −4 °C and stored at −80 °C until the analysis.

We have depicted the study overview, subjects, and analysis with a flow chart in Figure 1. All cohort members received periodic follow-ups based on the health information system (HIS) linked to cancer, stroke, death, and other vital report registry systems at different local disease control and prevention centers. In this study, the NSCLC cases were identified through the National Cancer Registry System implemented by hospitals in Shanghai. The information of local community residents was pooled by the Songjiang District Center for Disease Control and Prevention (Songjiang CDC) from the cancer registry data sources. The cancer-free controls were matched according to age and gender within the same district. A diagnosis of incident NSCLC was confirmed by the medical record reviewed by the staff from the department of non-communicable disease of the Songjiang CDC. In our study, the incident NSCLC cases were coded as C34.9 according to the *International Classification of Diseases, tenth revision* (ICD-10), and the information on the pathological diagnoses was collected from the cancer report system.

In this present study, the inclusion criteria of the NSCLC cases were: an incident of NSCLC and serum samples collected with sufficient volume without hemolysis; the exclusion criteria were: a participant with an alcohol-drinking habit, a prior diagnosis of another type of cancer at the baseline investigation, and being diagnosed with NSCLC after entering the cohort within a half year to exclude the subjects with preclinical stages of cancer. Participants in the cohort without having major health events such as a stroke, diabetes, cancer, and other chronic diseases self-reported in the baseline investigation were determined as being healthy, cancer-free participants. Among those, we randomly selected 40 subjects matched in age (±2 years) and gender as the cancer-free control group, but two serum samples were excluded because of hemolysis. Finally, 41 NSCLC cases and 38 cancer-free controls were included for further metabolomic profiling.

### 2.3. Data Collection and Measurement

At the baseline investigation, a structured questionnaire was conducted face-to-face by trained healthcare staff at the local community healthcare centers. Information on age, gender, education, socio-demographic information, lifestyles, and self-reported disease history was collected. Physical examinations, including height, weight, waist and hip circumference were carried out for every participant. Fasting blood was collected for a classical lipid profile, serum alanine aminotransferase (ALT), aspartate amino transferase (AST), and creatinine (the analysis was finished by Dian Diagnostic Co. Ltd. (Hangzhou, China) (a medical laboratory center). At follow-up, details of the NSCLC cases were extracted from the non-communicable disease department at the Songjiang CDC, which included the international classification (tenth revision) of diseases code, pathological diagnosis, the TNM classification of malignant tumor, etc.

### 2.4. Metabolomics Profiling and Data Preprocessing

The fasting serum samples were collected at the baseline investigation and were transported on ice and stored at −80 °C for each subject enrolled in our study. The serum samples were processed by the following steps: first, the frozen serum samples were thawed at 4 °C on ice. A total of 400 μL of the extraction solution (acetonitrile/methanol = 1/1, *v*/*v*) containing the isotopically-labelled internal standard mixture was added into an aliquot of serum samples (100 μL). Next, the samples were mixed on a vortex for 30 s, sonicated in the ice-watered bath for 10 min, and incubated for an hour to precipitate. Afterward, the samples were centrifuged at 12,000 rpm for 15 mins at 4 °C. Then, the supernatant was transferred to the glass vial for analysis through an ultra-high performance liquid chromatography-mass spectrometry (UPLC-MS) using the UHPLC system (Vanquish, Thermo Fisher Scientific) with a UPLC BEH Amide column coupled to a Q-Exactive HFX mass spectrometer (Orbitrap MS, Thermo). Details about the metabolite extraction and quality control process are presented in Appendix B. During the data acquisition process, one quality control (QC) sample was run after every ten samples. The LC-MS raw data were converted to the mzXML file using the ProteoWizard software (Pala Alto, California, USA). The peak detection, extraction, alignment, and integration were performed by the XCMS package (La Jolla, California, USA). The metabolite annotation was performed by the in-house MS2 database (BiotreeDB). The cut-off value for an annotation was set at 0.4.

Missing values were replaced with half of the minimum. A relative standard deviation of metabolites > 25% in the QC samples was considered unproducible and, thus, removed from further analysis. The data were normalized by dividing the peak area of the metabolite by the peak area of the internal standards, which is a common technique for untargeted metabolomics. We identified 3868 metabolite features (273 named and 3595 unnamed) under the positive ion mode of the Q-Exactive HFX mass spectrometer. After finishing the preprocessing, a data frame consisting of the retention time, mass-to-charge ratio values, and normalized peak intensity was subjected to further analysis.

### 2.5. Statistical Analysis

Continuous variables were presented as the mean ± (standard deviation [SD]) or median (interquartile range [IQR]). Categorical variables were expressed as a number (percentage). In univariate statistical comparisons, we used a chi-square or Fisher’s exact test for the categorical variables, a Student’s *t*-test or ANOVA for the normally distributed continuous variables, and a Wilcoxon signed-rank test or Kruskal–Wallis test for the skewed distributed continuous variables.

To minimize the impact of noise and the high variance of the variables, we scaled and logarithmically transformed the peak intensity. Then, the principal component analysis (PCA), an unsupervised analysis that could reduce the dimension of the data, was performed to visualize the distribution and grouping of the samples, including the QC samples. To visualize the separation of the groups, we performed a partial least squares-discriminant analysis (PLS-DA). The results for the PLS-DA with a positive ion mode are presented in Section 3.2. We performed an orthogonal projections to latent structures discriminant analysis (OPLS-DA) to find the significant different metabolites between the two groups. The metabolites with a variable importance in projection (VIP) > 1 were included for further analysis. The results of the PCA and OPLS-DA are presented in Appendix C. 

We applied the Benjamini–Hochberg procedure to control the false discovery rate (FDR). A *p*-value < 0.05 (using a *t*-test or Kruskal–Wallis test) and FDR corrected *p*-value (q-value) <= 0.2 were considered as significantly changed metabolites. We provide the named differential metabolites in the Appendix A: Part A. We set a strict selection criterion to select the differentially-expressed metabolites to generate the biologically valuable metabolites, given this study’s exploratory and hypothesis-generating nature. The selected 18 differentially-expressed metabolites were compared using the Student’s *t*-test to measure the relative abundance between the groups. The data were normalized by a log-transformation and UV scaling. A *p*-value ≤ 0.5 was considered statistically significant. The volcano plot of the overall metabolites identified across groups and the relative abundance of the differentially-expressed metabolites were presented in Section 3.2.

We used the R version 4.0.1 for the pathway analysis. The significantly changed metabolites (Appendix A) were mapped into their biochemical pathway using the Kyoto Encyclopedia of Genes and Genomes (KEGG) (https://www.kegg.jp) [13]. The pathway significance was determined based on the total number of metabolites mapped into the biochemical pathway and the total number of compounds in the involved pathway was calculated based on the hypergeometric distribution. The pathway was considered disturbed if the number of significant metabolites (i.e., hits) was ≥ 1 and the raw p of the hypergeometric distribution was < 0.10. The raw p was calculated based on the number of hits and the total number of compounds in a pathway. We used this lenient criterion because the sample size for each comparison group was small. The disturbed metabolic pathways enriched by the KEGG database are presented in Section 3.3. We provide the results of the KEGG enrichment analysis in Appendix A. Furthermore, a HCA was performed among the significantly changed metabolites in the Human Metabolome Database (HMDB) (https://hmdb.ca/) [14], using the “complete” method and the “Euclidean” distance measure. Finally, a Spearman correlation analysis was performed to evaluate the relationships between the significantly changed metabolites, baseline parameters, and classical lipids among all the participants.

We also performed a hierarchical clustering analysis (HCA) among the significantly expressed metabolites to identify whether types of metabolites tended to cluster together. In addition, we conducted a least absolute shrinkage and selection operator (LASSO) regression to optimize the variable selection and to identify whether any combinations of serum metabolite signatures could reduce the dimension in the data. Then, we applied a multivariable logistic regression analysis to assess the association between each metabolite and the risk of having NSCLC. The peak intensity of the metabolites was log-transformed in the regression analysis. Age and BMI were adjusted as continuous variables. Sex, education level, the family history of NSCLC, smoking status, drinking status, and exercise levels were retained as categorical covariates in the logistic regression models. Additionally, to assess the performance of the model, we conducted an internal validation with a three-fold cross validation, with the results presented in Section 3.4. Finally, we performed a Spearman correlation matrix between the significantly expressed metabolites, classical lipids, and baseline characteristics among all the subjects. The results are shown in Section 3.5. All the statistical analyses mentioned above were performed using the R version 4.0.1.

### 2.6. Role of the Funding Sponsors

The sponsors of the study declare no role in the study design, data collection, analysis, interpretation, or writing of the manuscript. The corresponding author had full access to all the data in this study and was responsible for the final decision to submit this paper for publication.

## 3. Results

### 3.1. Participant Characteristics

Table 1 presents the baseline characteristics of the study population. Of 79 participants, the NSCLC and the cancer-free controls were comparable in age and sex. The participants of our cohort primarily consisted of the elderly and middle-aged adults, leading to the majority of our study subjects being over 55 years old. 

### 3.2. Significantly Changed Metabolites and Their relative Changes across NSCLC Cases and Cancer-Free Controls

We first performed a PCA to assess the stability of the quality control (QC) samples and to visualize the metabolic separations across the NSCLC, cancer-free controls, and QC samples (Figure A1). Then, we analyzed the PLS-DA results and observed a significant separation between the NSCLC and cancer-free groups (Figure 2). The OPLS-DA model was applied to characterize the metabolic disturbances with the calculated VIP index. The OPLS-DA score plot and the result of the permutation test are shown in Figure A2. Afterward, we performed a Student’s *t*-test and Benjamini–Hochberg procedure to control the FDR due to multiple testing. Furthermore, we also calculated the fold change between the groups for each metabolite. In total, 3868 peaks were identified under the positive mode, with 273 named (the cut-off value for annotation was 0.4) and 3595 unnamed. The volcano plot of the overall characteristics of the metabolite peaks is shown in Figure 3. Using the selection criterion of a VIP > 1, *p*-value < 0.05 and corrected *p*-value (*q*-value) < 0.2, we identified 18 metabolites and considered them to be significantly-changed (or expressed) metabolites. Here, we conducted stepwise comparisons for the identified significantly-changed metabolites for identifying the common metabolic pathways underlying the cases with NSCLC. The relative abundance of 18 metabolites between the groups is shown in Figure 4.

### 3.3. Altered Metabolic Pathways across NSCLC Cases and Cancer-Free Controls

The significantly-changed metabolites were mapped into their biological metabolic pathways underlying NSCLC. The perturbed metabolic pathways are presented in Figure 5.

For the participants before a diagnosis with NSCLC with a median time of 1.44 (1.17–1.76) years, compared with the cancer-free controls, the choline metabolism in the cancer was the most significant disturbed pathway. Notably, the sphingolipid and glycerophospholipid (GPL) metabolisms were the metabolic disturbances between the two groups, with two hits matched in the pathways and with *p*-values of an enrichment analysis being 0.003 and 0.009, respectively. Additionally, apoptosis, linoleic acid and necroptosis were also the disturbed pathways with a *p*-value < 0.05 but with only one hit matched. The Sphingolipid signaling pathway, nicotinate and nicotinamide, and the retrograde endocannabinoid signaling were the matched pathways with a *p*-value < 0.1, but they did not meet the criteria for a disturbed metabolic pathway, possibly due to the small sample size.

### 3.4. Serum Metabolite Signatures for NSCLC

We explored whether any combinations of the serum metabolites could reduce the dimension in the data and serve as signatures of NSCLC. A hierarchical clustering analysis (HCA) was based on the 18 differentially-expressed metabolites, which provided intuitive visualizations of the trends in metabolites between the two groups (Figure 6a). We focused on the clusters that showed an upward or downward trend between the NSCLC and cancer-free controls. The first composition of 12 metabolites (mainly lipids and lipid-like molecules, organic acids and derivatives, and organic nitrogen compounds) showed a decreasing trend. The second composition of three metabolites (mainly lipids and lipid-like molecules and organoheterocyclic compounds) showed an increasing trend.

We then performed a LASSO regression to select the variables that could be the optimal indicators of the present risk factors, including the baseline characteristics and differentially-expressed metabolites. We identified a set of metabolites including five metabolites that could serve as the serum metabolite signatures of NSCLC: N1-Methyl-4-pyridone-3-carboxamide (organoheterocyclic compounds), 2,6-Dimethoxy-4-(1-proponyl) phenol (benzenoids), tigloidine (alkaloids and derivatives), artonin C (phenylpropanoids and polyketides), and pipericine (lipids and lipid-like molecules). The logistic regression model built by these five predictors displayed a high ability with the area under the ROC curve (AUC) of 0.94 (a 95% confidence interval: 0.89–0.98) in our dataset. Moreover, by adding the smoking habits, the BMI and TG levels into the model, the model showed a maximum ability with the AUC of 0.99 (a 95% CI: 0.98–1.00). In conclusion, the serum metabolite signatures, BMI, TG, and smoking habits could precisely predict the participants with NSCLC in our study. The results are shown in Figure 6b. In addition, due to the small sample size, we performed an internal three-fold cross validation to assess the performance of the risk prediction model. The results are presented in Figure 6. We found that the mean AUC = 0.90 for the ROC with a three-fold cross validation of the model based on the metabolite signatures, BMI, TG, and smoking habits (in Figure 7a). The mean AUC = 0.84 for the ROC with a three-fold cross validation of the model based only on the metabolite signatures (in Figure 7b). Although the internal cross-validation showed a good performance of our risk prediction models based on the metabolite signatures and baseline characteristics, a future external validation is warranted.

### 3.5. Correlation between Significantly Changed Metabolites, Baseline Characteristics, and Classical Lipids

We conducted a Spearman correlation matrix between the significantly-changed metabolites, baseline clinical characteristics, and classical lipid levels among all the participants (Figure 8). The metabolites that increased or decreased tended to cluster together in the correlation matrix. The body mass index (BMI), waist-to-hip ratio, triglyceride (TG), and years since a NSCLC diagnosis, were clustered together with the upward metabolites including N1-methyl-pytidone-3-carboxamide, Sphingosine, L-Palmitoylcarnitine, Hexacosanoyl carnitine, and Artonin C. The total cholesterol (TC), high-density lipoprotein (HDL) cholesterol, and low-density lipoprotein (LDL) cholesterol were clustered together with the downward metabolites, e.g., mostly the lipids and lipid-like molecules, organic acids and derivatives, and organic nitrogen compounds. In addition, age and second-hand exposure were clustered together with tigloidine, the alkaloid and derivative, which is the downward-expressed metabolite.

## 4. Discussion

To our knowledge, we report the first comprehensive serum metabolomics characteristics of NSCLC patients before the time of a disease diagnosis among Chinese community residents. In the PLS-DA, we observed that the first and second component scores were relatively low due to the possible reasons that the blood serum metabolomics data were highly multi-dimensional with high noise, while the sample size in our study was relatively small. However, we observed that the group separation was optimized with the PLS-DA compared with the PCA. We further performed an OPLS-DA to assist in identifying important metabolites. Using a strict standard (VIP > 1, a *p*-value < 0.05 and a *q*-value ≤ 0.2), we found 18 metabolites significantly-expressed between the groups after accounting for multiple comparisons. Among them, two were positively-associated with a risk of NSCLC and showed an upward trend in the NSCLC group, while four had negative associations and showed downward trends. We performed a LASSO regression and identified that N1-Methyl-4-pyridone-3-carboxamide, 2,6-Dimethoxy-4-(1-proponyl) phenol, tigloidine, artonin C, and pipericine could serve as the serum metabolite signatures of NSCLC with the AUC reaching 0.94 (a 95% CI: 0.89–0.98). Metabolic alterations identified in the KEGG pathway enrichment analysis imply that pathways including the choline metabolism, sphingolipid and glycerophospholipid metabolism, linoleic acid metabolism, and cell apoptosis and necroptosis were associated with a NSCLC risk before diagnosis.

Metabolic reprogramming is one of the hallmarks of cancer. Tumoral cells exhibit significant metabolic alterations and change their capability to metabolize carbohydrates, lipids and proteins to support cell proliferation [15]. The abnormality of lung tumors occurs in vessel structures that could limit the nutrient supply to the tumoral cells and produce hypoxia, which could induce multiple metabolic alterations.

In our study, we identified that D-1-Amino-2-pyrrolidinecarboxylic acid, a compound that belongs to proline and its derivatives, was significantly lowered in the NSCLC group. Proline has a unique role in metabolic regulation. Recent research has shown that proline metabolism plays a critical role in cancer metabolic reprogramming [16,17,18]. Proline has its α-amino group within a pyrrolidine ring and, thus, it is the sole proteinogenic secondary amino acid with its own metabolic pathways [12]. Proline dehydrogenase/proline oxidase (PRODH/POX) is encoded by tumor protein 53 (TP-53)-induced gene 6 (PIG-6), while TP-53 is the most frequently mutated gene in NSCLC [19,20]. In cancer cells, proline functions as a source for cancer cellular energy production and as an intermediate between the tricarboxylic acid (TCA) cycle and the urea cycle [21]. Researchers have already identified the critical role of proline catabolism in NSCLC. PRODH is activated to reduce the proline levels by the chromatin remodeling factor lymphoid-specific helicase (LSH), an epigenetic driver of NSCLC. PRODH promotes NSCLC tumorigenesis and could promote the expression of inflammatory genes and tumor cell proliferation [22]. In our study, the relative abundance of (3xi,6xi)-Cyclo(alanylvalyl) was significantly lowered in the NSCLC group. Moreover, (3xi,6xi)-Cyclo(alanylvalyl) belongs to the class of organic compounds known as alpha amino acids and derivatives and amino acids, such as methionine and tryptophan, have been reported to be associated with lung cancer [23]. Methionine is found to be involved in nucleotide biosynthesis via the one-carbon metabolism pathway and protein synthesis, which are the critical activities in cancer cells [24]. Additionally, methionine is also involved in the formation of glutathione, a biomarker of oxidative stress and it might contribute to chronic inflammation and cancer development [23].

We found that the relative concentration of N1-Methy-4-pyridone-3-carboxamide was significantly higher among the NSCLC group. N1-Methyl-4-pyridone-3-carboxamide belongs to the class of organic compounds known as nicotinamides. Nicotinamide (NAM) is the water-soluble form of Vitamin B3 (niacin) and a precursor of nicotinamide-adenine dinucleotide (NAD^+^), which takes part in several redox and non-redox reactions that regulates cellular energy metabolism [25]. The uncontrolled proliferation of cancer cells requires the reprogramming of their metabolism, which is critical for their survival and growth. In dehydrogenase reactions, NAD^+^ acts as a co-enzyme to produce adenosine triphosphate (ATP). High NAD+ could suppress the production of reactive oxygen species (ROS) and increase the mitochondrial quality; thus, it could protect against oxidative stress and improve cell survival [26]. In addition, we also identified nine lipids and lipid-like molecules that were significantly-expressed across the groups. Lipid metabolism is highly altered among proliferating cells because cancer cells increase de novo adipogenesis for membrane biosynthesis and signaling molecules instead of relying on the uptake of exogenous fatty acids (FA). We identified four types of phosphatidylcholine (PC) species. Phospholipids are the primary component of cell membranes and work as the second messengers in intracellular signal transduction. Researchers have found that the phospholipids were altered in NSCLC. For example, Marien et al. used 167 NSCLC patients’ tissue samples and profiled 179 phospholipids species that were altered between malignant and matched controls by a mass spectrometry-based approach. Their most important findings included a decrease in the sphingomyelins (SM) and an increase in several phosphatidylethanolamines (PEs) and PC species [27]. In our study, we also found phospholipid alterations among the NSCLC group; however, phospholipid alterations have also been described in other tumor types such as lymphoma [28]. Changes in phospholipid metabolism are not distinctive in NSCLC because it occurs early in carcinogenesis, irrespective of the cancer subtype and cancer stage [15].

Sphingosine (Sph), a metabolite of sphingomyelin, was over-expressed in the NSCLC group within our study. The metabolites of sphingomyelin, such as Sph, ceramide (Cer) and sphingosine-1-phosphate (S1P), form the lipid bilayers of cell membranes and play critical roles in the development of cancer [29]. Sph and Cer induce cell apoptosis, while S1P promotes cancer cell growth, survival, angiogenesis, and inflammation [30]. When the physiological condition is normal, the expression of Sph, Cer, and S1P should maintain a dynamic balance through enzymatic reactions that are described as forming a ‘sphingolipid-rheostat’ that is crucial for cell survival [31]. Sphingosine kinases (SPHKs) could phosphorylate Sph to S1P, which are the critical mediators of the ‘sphingolipid-rheostat.’ SPHK1 and SPHK2 are the two isoenzymes of sphingosine kinase, while SPHK1 has been reported as an effective pharmacologic target in anticancer therapy by inhibiting its activity of promoting the transition from Sph/Cer to S1P [32]. Song et al. reported that the expression of SPHK1 was markedly increased in NSCLC and associated with cancer progression and a poor survival of patients with NSCLC. The upregulation of SPHK1 significantly inhibited apoptosis, it was associated with the induction of antiapoptotic proteins, and it promoted the proliferation and migration of NSCLC cells in vitro and in vivo. In contrast, silencing SPHK1 expression can induce the apoptosis of NSCLC cells and increase the chemosensitivity of NSCLC to cytotoxic drugs [33]. Our results further proved that the metabolic balance of sphingolipids is of paramount importance in the cellular activities in NSCLC, which could be detected in the prior-to-diagnosis serum samples. In conjunction with the baseline characteristics, the metabolites identified by the LC-MS techniques could be used as a less-invasive, early disease detection tool among Chinese community residents.

Previous findings have showed that the metabolic pathways altered in lung cancer patients included those of glycolysis (the “Warburg effect”), lipids, choline phospholipid (the Kennedy pathway), one-carbon, amino acids, nucleotide metabolism, oxidative stress, and inflammation [34,35]. Consistent with the previous literature, our study proved that NSCLC patients notably presented alterations in the choline metabolism compared with the cancer-free participants. The GPL metabolism was also one disturbed pathway presented in comparisons between the NSCLC and cancer-free participants, followed by the sphingolipid metabolism. Moreover, we identified metabolomic characteristics specific for NSCLC prior to diagnosis that reflected 13 metabolites whose relative abundance showed a downward trend (mainly lipids and lipid-like molecules, organic acids and derivatives, organic nitrogen compounds, alkaloids and derivatives, and benzenoids). Additionally, five metabolites showed an upward trend (mainly the organoheterocyclic compound, phenylpropanoid and polyketide, lipids and lipid-like molecules, and the organic nitrogen compound). Among these 18 significantly-expressed metabolites, besides the metabolites involved in the KEGG metabolic pathway, we also identified a few metabolites, such as (3xi,6xi)-Cyclo(alanylvalyl) and Na,Na-Dimethylhistamine, that were the potential biomarkers for the consumption of some specific foods. Moreover, a Spearman correlation matrix further indicated that the metabolites that showed an upward or downward trend tended to cluster together across the groups and related to each other with the baseline characteristics in the expected direction.

In cancer, metabolomics detects oncological developments through measurable metabolic profiles from the metabolic pathways through global metabolite variations [36]. A growing body of literature shows that abnormal choline metabolism is a distinguishing feature of carcinogenesis and tumor progression [37,38]. In the NSCLC group, we identified four types of phosphatidylcholine (PC), such as PC (22:6/20:3), PC (20:4/20:3), and PC (22:6/20:4), that showed a downward trend. Phosphatidylcholine (PC) and other phospholipids such as phosphatidylethanolamine (PE) are neutral lipids, forming the bilayer structure of cellular membranes and regulating membrane integrity, and they are the most abundant phospholipids in the cell membrane [39]. To synthesize PC and PE, cytidine diphosphate (CDP)-choline and CDP-ethanolamine are required. The biosynthesis and hydrolysis of PC could mediate the mitogenic signal transduction events in cells. Phosphocholine (PCho), diacylglycerol (DAG), and arachidonic acid metabolites, which are the products of choline phospholipid metabolism, might function as second messengers that are important to the mitogenic activity [40,41]. Growth factor stimulation, cytokines, oncogenes, or the requirements for eicosanoid production also regulate choline phospholipid metabolism [42,43,44]. A network of transporter systems and enzymes are involved in the choline phospholipid metabolism that is deregulated in cancer cells. In addition, the tumor microenvironment also affects choline metabolism. Hypoxia and an acidic extracellular pH could lead to the abnormal physiological environments caused by tumors. Studies have found that an acidic extracellular pH could significantly increase the glycerophosphocholine (GPC) level and decrease the PCho levels in perfused mammalian cells [45].

Sphingolipid metabolism was also a significantly altered pathway between the groups, apart from the choline metabolism. Sphingolipids, the structural molecules of cell membranes, play an important role in regulating cancer cell death and survival by controlling the cell-signaling functions. Our data showed that the critical metabolites involved in this pathway were sphingosine and glucosylceramide, and the abundance was significantly different between the groups. In the NSCLC group, the relative abundance of sphingosine was significantly higher than the controls, while glucosylceramide showed a downward trend. In cells, cellular stress could induce the generation of sphingosine and/or ceramide to mediate cancer cell death by activating the de novo synthesis pathways, sphingomyelin hydrolysis or the salvage pathway, which could explain why we observed an increased abundance of sphingosine in NSCLC. Many tumors could increase the ceramide metabolism by increasing the activities of glucosyl-ceramide synthase (GCS), sphingomyelin synthase (SMS), and ceramide kinase (CERK), etc., which could increase the generation of sphingolipids with pro-survival functions [46]. In addition, the accumulation of ceramide in response to cellular stress could mediate cancer cell death through the induction of apoptosis, necroptosis, autophagy, and ER stress. Endogenously generated ceramide is an inducer of apoptosis regulated by various mechanisms in a cell-type-dependent and/or context-dependent manner [47,48,49]. In contrast, there are studies that have demonstrated that ceramide could protect some cancer cells from cell death [50,51]. Overall, ceramide mediates apoptosis in most cancer cells via mitochondrial membrane perturbation and/or influencing the cell-death signaling; however, it might have protective effects against apoptosis depending on the downstream ceramide targets, the subcellular localization of ceramide and the type of stress stimuli. Our data showed that aberrant sphingolipid metabolism with an abnormal accumulation of sphingolipids and ceramides could be observed among the Chinese residents’ serum samples before their diagnosis.

In our study, we found four types of PC that showed a downward trend in the NSCLC group when compared with the cancer-free controls. Underlying these phenotypic alterations in the choline metabolism might indicate the degradation of PC, resulting in the dysplasia of the surrounding substances of the cell membrane. Same as with the previous findings, the aberrant expression of PC or its related compounds could be detected by a non-invasive LC-MS to detect NSCLC at an early stage; however, abnormal choline phospholipid metabolism has also been observed in many other types of cancer, not heterogeneous to the NSCLC [38,52]. We observed that the AUC of the serum metabolite signatures combined with the smoking history, BMI, and TG indicators for identifying NSCLC could reach 0.99 (a 95% CI: 0.98-1.00). This proved that the serum metabolite fingerprints combined with the baseline characteristics could be an effective tool to identify an NSCLC presence within our study. However, future validations are warranted to test whether our results could be applicable to populations from other districts in China.

Several research groups have recently applied metabolomic techniques to unveil the metabolic alterations associated with lung cancer, with most of the studies being case-control studies. Hori et al. detected 58 metabolites in serum using chromatography/mass (GC/MS) and found 23 differentially-expressed metabolites in the Japanese population [8]. Maeda et al. studied 21 amino acids in NSCLC patients’ plasma with LC/MS and proved that the amino acid profiles could be used for screening NSCLC among the Japanese population [53]. Jordan et al. applied a nuclear magnetic resonance (NMR) to measure 21 metabolites and showed that serum metabolomics could differentiate lung cancer between patients and healthy controls [54]. Neither the study design nor the sample population was different to ours. To avoid reverse causation, we used a prospective nested case-control study to illustrate the serum metabolomic alterations before diagnosing NSCLC. Schult TA et al. established the magnetic resonance spectroscopy (MRS)-based metabolomics predictive models to identify the lung cancer presence in the U.S. population [7]. However, serum metabolomic profiling prior to a diagnosis with NSCLC is scarce in China and, thus, the metabolomic predictive models could not be established. Our findings could have implications for lung cancer screening in China. Considering that the metabolite fingerprints might improve the identifying of participants with the highest risk of NSCLC, who could benefit the most from the screening, a metabolomic predictive tool validated in other cohorts is warranted in future studies. The public health implication of this study is the importance and the feasibility of the non-invasive and early-detection of NSCLC with LC-MS among the Chinese community residents. Healthcare centers throughout the nation could evaluate the use of spectroscopic imaging to assist in preventing and diagnosing NSCLC. Additionally, exploring choline phospholipid metabolism and sphingolipid metabolism might help to identify new therapeutic targets for NSCLC treatments. Because the LC-MS technique is relatively expensive and qualified practitioners are in short supply, the cost-effectiveness of its ability for early-detecting NSCLC could be evaluated before being widely applicated.

The strength of our study is this prospective setting. The reverse causation could be mitigated with our study design; however, several limitations should be considered. First, we did not validate the results of the identified metabolites with an untargeted LC-MS technique to reconfirm the robustness of the metabolites that we measured. Although we applied a standard LC-MS data acquisition operation to identify the significantly-expressed metabolites, the validation of the results should be paramount and must be improved in our future studies. Second, from the perspective of statistics, the sample sizes for the NSCLC and cancer-free controls were relatively small. Third, although we identified metabolomic alterations before a diagnosis with NSCLC among our study participants, and conducted an internal cross validation of our risk prediction model, the extrapolation has not been established through external validation in other participants from our cohort and this should be improved in future research. Last but not least, there is a possibility that the results could be affected by other confounders that we did not consider. For example, the serum metabolite profiles could have been affected by diet composition and circadian rhythm. We did identify a few metabolites that might be the potential biomarkers for the consumption of certain foods; however, we did not assess the impact of dietary composition in our study, which should be improved in our future research.

## 5. Conclusions

In summary, we reported the metabolome profile of NSCLC patient blood serum samples collected before disease diagnosis and compared the results with those observed in serum samples obtained from healthy control subjects. We identified 18 significantly-expressed metabolites, with most of them being lipid and lipid-like molecules, organic acids, and nitrogen compounds. Additionally, choline metabolism exhibits a markedly metabolic disturbance in NSCLC patients, followed by sphingolipid metabolism and glycerophospholipid metabolism. Our data point to the metabolisms mentioned above and the related metabolites as the potential critical targets for cancer interventions that could help with the early diagnosis of NSCLC patients. Our results demonstrate the potential feasibility of applying LC-MS-based serum metabolomics to predict the presence of NSCLC combined with baseline characteristics, such as smoking habits and the exposure history to second-hand smoking. With future validation, metabolomic techniques might be used as an effective and less-invasive tool to supplement the screening programs of NSCLC in the Chinese population.

## Figures and Tables

**Figure 1 metabolites-12-00906-f001:**
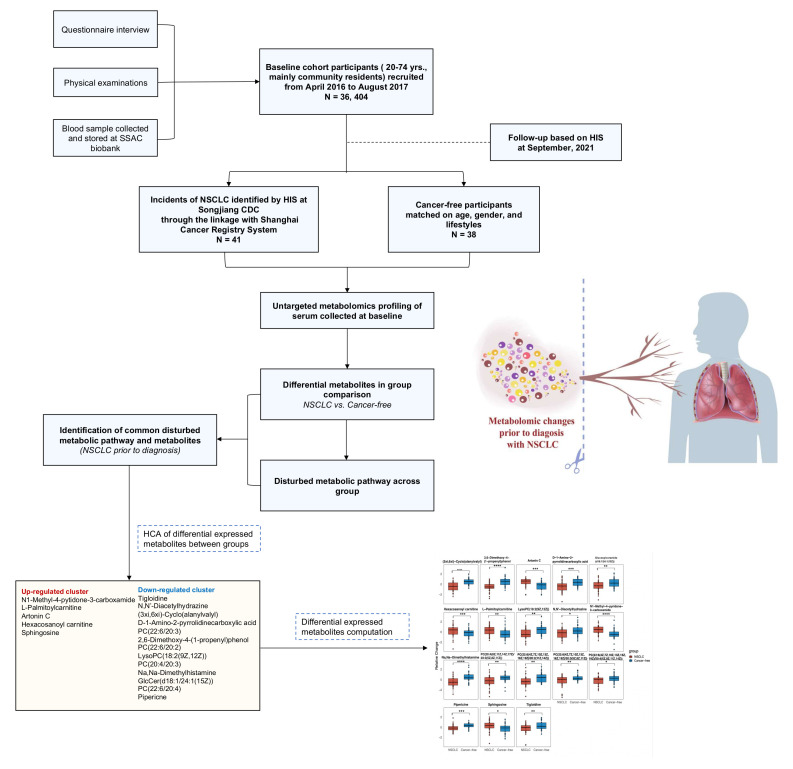
Flow chart of the study overview, subjects, and analysis.

**Figure 2 metabolites-12-00906-f002:**
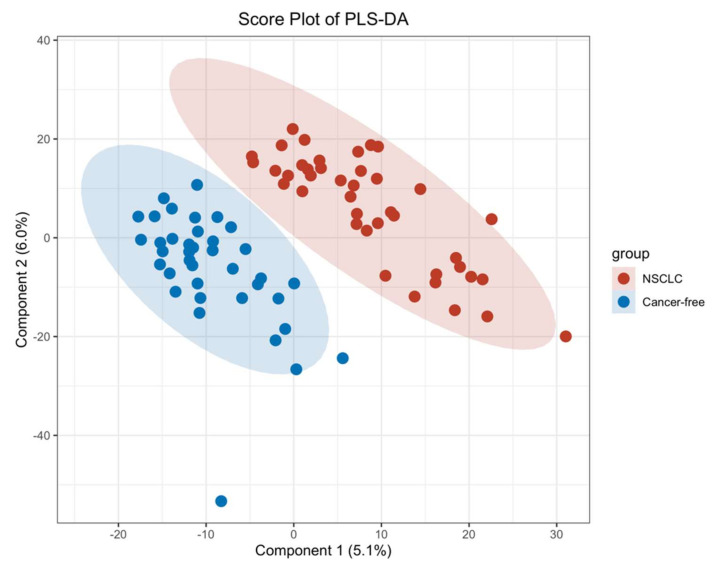
PLS–DA score plot between the NSCLC group and cancer–free control group.

**Figure 3 metabolites-12-00906-f003:**
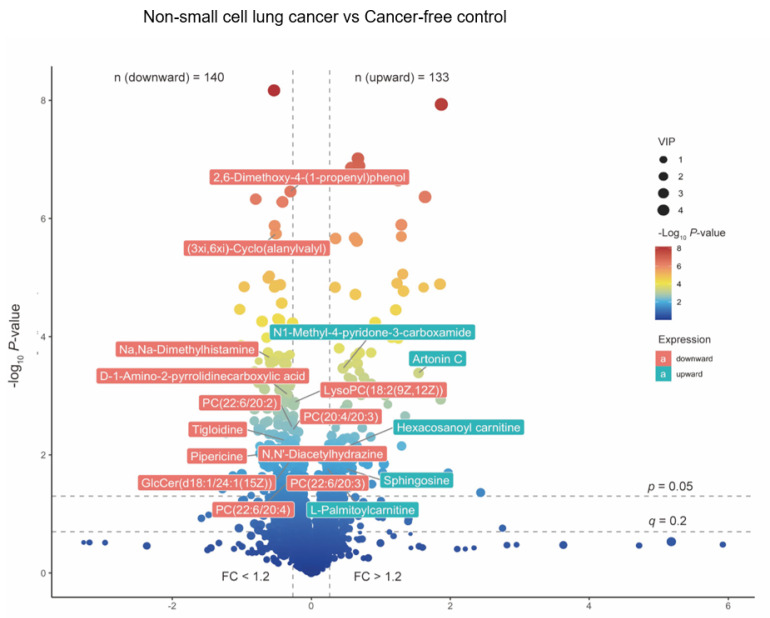
The volcano plot of overall metabolites identified between NSCLC cases and cancer–free controls. The *p*-value was calculated based on Student‘s t-test or Kruskal–Wallis test; The q-value was calculated with Benjamini–Hochberg procedure to control the FDR. FC was calculated based on the relative abundance across groups. The 18 significantly altered metabolites were labeled. Red labels stand for NSCLC significantly depleted metabolites, and green labels stand for NSCLC significantly enriched metabolites. Abbreviations: Lysophospholipid: LysoPC; PC: Phosphatidylcholine; GlcCer: Glucosylceramide.

**Figure 4 metabolites-12-00906-f004:**
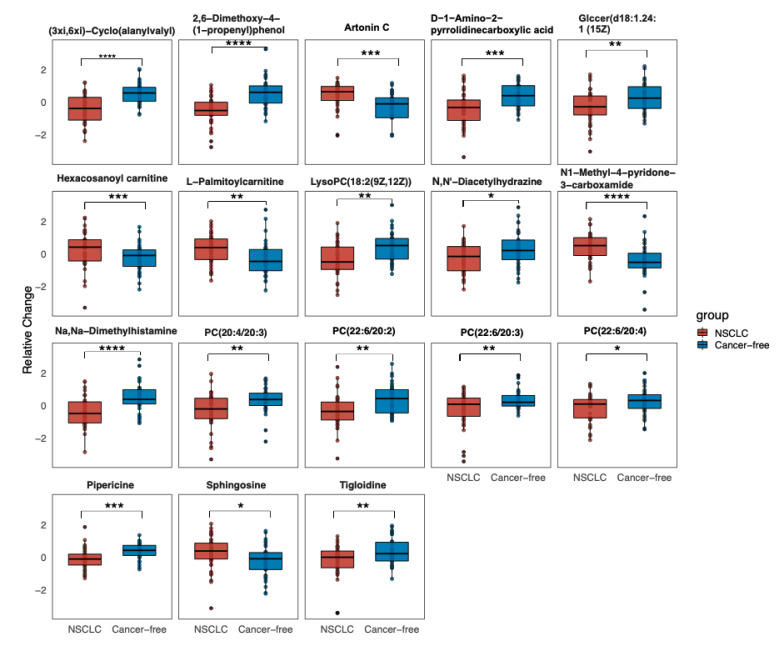
Relative abundance of 18 significantly changed metabolites across the groups (the relative expressed values were compared using a Student’s *t*-test). Data are the normalized values by log-transformation and UV scaling. The presented box shows the 25th and 75th percentiles. The horizontal line represents the mean values of each metabolite. *p* < 0.05, *; 0.001 < *p* < 0.01, **; 0.0001 < *p* < 0.001, ***; *p* < 0.0001, ****. Abbreviation: GlcCer: Glucosylceramide; PC: Phosphatidylcholine; LysoPC: Lysophospholipid.

**Figure 5 metabolites-12-00906-f005:**
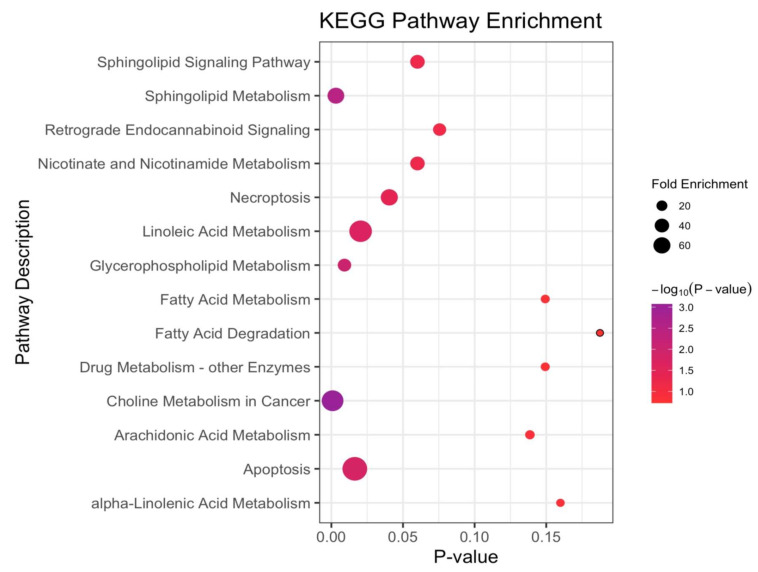
The disturbed metabolic pathways when comparing the NSCLC cases versus cancer-free controls. Various metabolism changes before a diagnosis with NSCLC was manifested (the *p*-value was calculated based on the hypergeometric distribution; fold enrichment was calculated based on the total number of metabolites mapped into the biochemical pathway; and the total number of compounds in the involved pathway).

**Figure 6 metabolites-12-00906-f006:**
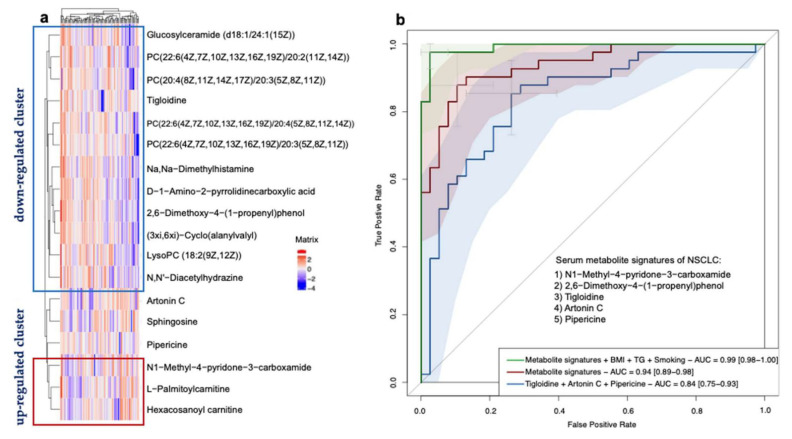
Identification of NSCLC-associated serum metabolite signatures. (**a**) Hierarchical clustering analysis of 18 significantly differentially-expressed metabolites according to OPLS-DA VIP values. The definition of significantly-expressed metabolites was the same as in Figure 4. (**b**) Receiver operating characteristic curve based on the logistic regression of metabolites selected from the LASSO regression and characteristics from the baseline investigation to identify the metabolite signatures and assess their probabilities in identifying NSCLC. The x-axis represents the false-positive rate of the risk prediction. The y-axis represents the true-positive rate of the risk prediction. The lines represent the performance of the model, and the shades represent the confidence intervals of the area under the curve.

**Figure 7 metabolites-12-00906-f007:**
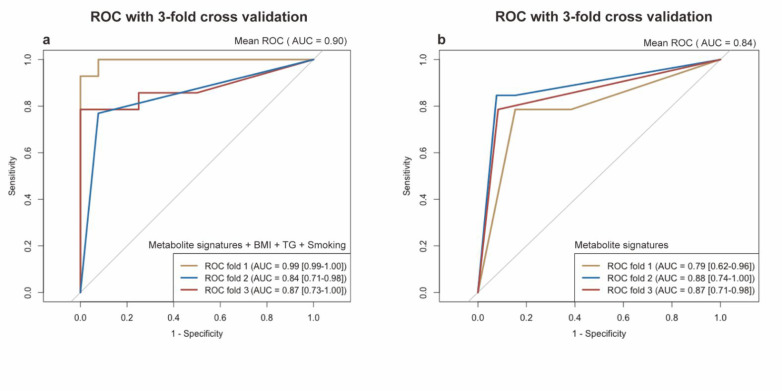
Receiver operating characteristic (ROC) curve with internal 3-fold cross validation. (**a**) ROC curve of logistic model based on the metabolite signatures, e.g., BMI, TG, and smoking. (**b**) ROC curve of logistic model only based on the metabolite signatures. The x-axis represents the false-positive rate. The y-axis represents the true-positive rate. The lines represent the performance of each fold. Metabolite signatures include N1-Methyl-4-pyridone-3-carboxamide, 2,6-Dimethoxy-4-(1-propenyl)phenol, Tigloidine, Artonin C, and Pipericine.

**Figure 8 metabolites-12-00906-f008:**
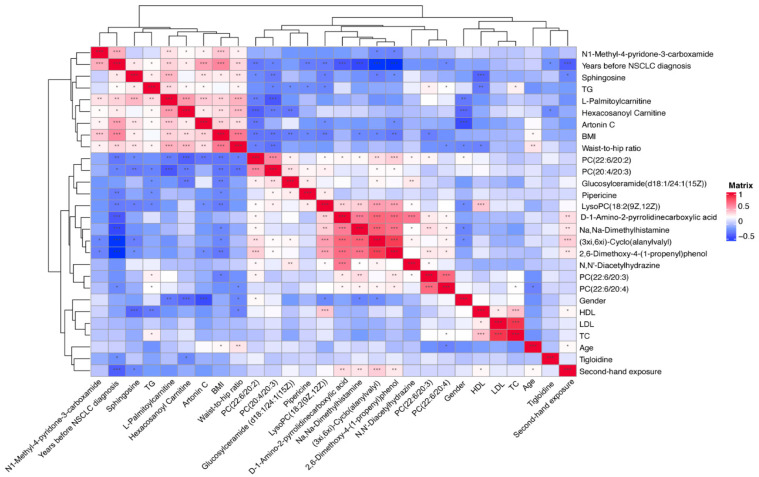
Spearman correlation matrix between significantly changed metabolites, classical lipids, and baseline characteristics in all subjects. Shades of blue represent a negative correlation coefficient; shades of red represent a positive correlation coefficient. Variables were ordered using a hierarchical clustering analysis. Threshold for significance testing of the Spearman correlation: 0.01 < *p* < 0.05, *; 0.001 < *p* < 0.01, **; *p* < 0.001, ***.

**Table 1 metabolites-12-00906-t001:** Characteristics of the study participants.

Characteristic	NSCLC (*n* = 41)	Cancer-Free Control (*n* = 38)	Total (*n* = 79)	*p*-Value
**Age, years**				0.371
41–55	9 (21.95%)	8 (21.05%)	17 (21.52%)	
56–65	18 (43.90%)	22 (57.89%)	40 (50.63%)	
65–75	14 (34.15%)	8 (21.05%)	22 (27.85%)	
Mean ± SD (years)	61.24 ± 6.97	60.21 ± 6.74	60.75 ± 6.84	0.505
**Gender**				0.411
Male	21 (51.22%)	15 (39.47%)	36 (45.57%)	
Female	20 (48.78%)	23 (60.53%)	43 (54.43%)	
**Education**				0.558
Middle school or below	20 (48.78%)	22 (57.89%)	42 (53.16%)	
High school or above	21 (51.22%)	16 (42.11%)	37 (46.84%)	
**History of Respiratory Diseases**				0.241
Yes	3 (7.32%)	0 (0.00%)	3 (3.80%)	
No	38 (92.68%)	38 (100.00%)	76 (96.20%)	
**Smoking Status**				0.160
Never	27 (65.85%)	31 (81.58%)	58 (73.42%)	
Former	1 (2.44%)	0 (0.00%)	1 (1.27%)	
Current	13 (31.71%)	7 (18.42%)	20 (25.32%)	
**Second-hand Exposure**				0.026
Yes	6 (14.63%)	0 (0.00%)	6 (7.59%)	
No	35 (85.37%)	38 (100.00%)	73 (92.41%)	
**Alcohol drinking**				0.228
Yes	0 (0.00%)	2 (5.26%)	2 (2.53%)	
No	41 (100.00%)	36 (94.74%)	77 (97.47%)	
**Exercise**				0.185
Yes	14 (34.15%)	7 (18.42%)	21 (26.58%)	
No	27 (65.85%)	31 (81.58%)	58 (73.42%)	
**BMI**, ***kg*****/*m*^2^**	25.08 (22.88–27.29)	21.73 (20.72– 22.55)	22.81(21.35–25.28)	<0.001
**Waist to hip circumference ratio**	0.88 ± 0.06	0.85 ± 0.05	0.88 ± 0.06	<0.001
**HDL cholesterol, mmol/L**	1.31 (1.12–1.56)	1.40 (1.31–1.49)	1.40 (1.21–1.52)	0.156
**LDL cholesterol, mmol/L**	2.8 (2.27–3.28)	2.78 (2.50–3.12)	2.79 (2.39–3.12)	0.910
**TG, mmol/L**	1.4 (1.17–1.76)	1.16 (0.90–1.79)	1.29 (1.07–1.79)	0.016
**TC, mmol/L**	4.87 ± 1.12	4.76 ± 0.57	4.82 ± 0.90	0.591
**Time to diagnosis**	1.44 (1.17–1.76)	NA	1.44 (1.17–1.76)	
**Histological subtypes**				
**Adenocarcinoma**	32 (78.05%)	NA	32 (78.05%)	
**Squamous cell carcinoma**	4 (9.76%)	NA	4 (9.76%)	
**Other subtypes**	5 (12.20%)	NA	5 (12.20%)	

Values are mean ± standard deviation (SD) for normally-distributed variables, median (interquartile range, (IQR)) for skewed-distributed variables, or n (%) for categorical variables. Abbreviations: BMI: body mass index; HDL: high-density lipo-protein; LDL: low-density lipo-protein; TG: triglyceride; TC: total cholesterol; NA: not applicable. *p*-values were calculated from a Wilcoxon signed-rank test for skewed continuous variables, a Student’s *t*-test for variables with normal distributions, and a Chi-square test or Fisher’s exact test for categorical variables.

## Data Availability

Raw data in this study are available from the corresponding author upon reasonable request. All the data that support the findings of this study are free to obtain in the paper or in the supplementary materials.

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
