# Peer review of "Serum Metabolomics Profiling Reveals Metabolic Alterations Prior to a Diagnosis with Non-Small Cell Lung Cancer among Chinese Community Residents: A Prospective Nested Case-Control Study"

_metabolites, 2022, doi:10.3390/metabo12100906_

Round 1
Reviewer 1 Report
This article described the research of serum metabolome to diagnose non-small cell lung cancer (NSCLC). Since early detection increases the survival ratio, it is crucial to find any minor signs of NSCLC. The studies and data processing are very well planned in this article, and the results are clear. The authors identified 18 significantly expressed metabolites: the patients' expression differs from cancer-free control groups. The number of cases is enough to reach a conclusion. Their LC-MS-based serum metabolomics in conjugation with baseline characteristics will be the key to developing a new diagnostic procedure. The results are valuable since this manuscript contains helpful information for further developments in this area. I recommend this manuscript to publish in Metabolites with the following minor revision.
In Table 1, characteristics (Age, Gender, Education, etc.) are better to be highlighted or bold to make them separate from the categories. Or Characteristics in a different column, separating each characteristic's classes.
Author Response
Dear reviewer,
Thank you for your careful review of our work and for the opportunity to revise our manuscript. We have revised the manuscript according to your suggestions. The elaboration below was made point by point and all the revisions were marked with Track Change with the number of lines in our revised manuscript.
Point 1: This article described the research of serum metabolome to diagnose non-small cell lung cancer (NSCLC). Since early detection increases the survival ratio, it is crucial to find any minor signs of NSCLC. The studies and data processing are very well planned in this article, and the results are clear. The authors identified 18 significantly expressed metabolites: the patients' expression differs from cancer-free control groups. The number of cases is enough to reach a conclusion. Their LC-MS-based serum metabolomics in conjugation with baseline characteristics will be the key to developing a new diagnostic procedure. The results are valuable since this manuscript contains helpful information for further developments in this area. I recommend this manuscript to publish in Metabolites with the following minor revision.
Response 1: We truly appreciate the Reviewer’s positive comments.
Point 2: In Table 1, characteristics (Age, Gender, Education, etc.) are better to be highlighted or bold to make them separate from the categories. Or Characteristics in a different column, separating each characteristic's classes.
Response 2: We set the characteristics with bold type (in table 1, page 7 and page 8).
We are truly appreciated for your suggestions!
We believe that the manuscript has been improved considerably and hope you will find it suitable for publication in Metabolites. However, we are willing to make further changes if needed.
Sincerely,
Genming Zhao, MD, PhD
Professor of Epidemiology
Fudan University School of Public Health
131 Dong An Road, Shanghai 200032, PRC
Tel: 86-21-54237334
E-mail: gmzhao@shmu.edu.cn
Reviewer 2 Report
This manuscript examined the metabolomic alterations in serum of non-small cell lung cancer (NSCLC) patients among Chinese residents. The measurements of changes in biomarkers are conducted through the LC-MS based metabolomic analysis, and correlate these biomarkers with baseline characteristics. The results identified related biomarkers and associated pathways.
Line 382: States D-1-Amino-2-pyrrolidinecarboxylic acid was significantly expressed in NSCLC group. But in Figure 4, level of this metabolite was lower in NSCLC compared to cancer free samples.
For those identified metabolites, are they confirmed by authentic standards or MSMS data?
Any explanation on why 1st and 2nd component scores are pretty low in Figure 2?
Author Response
Dear reviewer,
Thank you for your careful review of our work and for the opportunity to revise our manuscript. We now revised our manuscript according to your comments. Please kindly check the attachment.
Best regards,
XIANG Yu

Round 2
Reviewer 2 Report
Thank you for your detailed explanations. The confused sentences were addressed properly. I do not have any other comments for this manuscript. Overall, it is good to go.